# Low-Temperature-Induced Changes in Rice Panicle Architectures and Their Robustness in Extremely Cold-Tolerant Cultivars

**DOI:** 10.3390/plants14172759

**Published:** 2025-09-03

**Authors:** Masato Kisara, Aisha Ahmad Abu, Atsushi Higashitani

**Affiliations:** 1Miyagi Prefectural Furukawa Agricultural Experiment Station, 88 Fukoku, Furukawa, Osaki 989-6227, Miyagi, Japan; 2Graduate School of Life Sciences, Tohoku University, 2-1-1 Katahira, Aoba-ku, Sendai 980-8577, Miyagi, Japan; aishaahmad1714@gmail.com

**Keywords:** basal branch, cold tolerance, LT stress, *Oryza sativa* L., panicle architecture, spikelet

## Abstract

Low-temperature (LT) stress remains a challenge in rice cultivation and breeding. Despite global warming, cold waves cause damage to rice plants, particularly during pollen development. LTs during early panicle formation worsen pollen formation defects, but the underlying mechanisms remain unclear. We investigated the effects of low temperatures (19.0 °C and 18.5 °C) throughout reproductive growth on the panicle architecture and fertility of 28 *japonica* rice varieties with different LT tolerances. LT-sensitive varieties like Sasanishiki and conventional LT-tolerant varieties like Hitomebore showed increased spikelet densities on basal branches, whereas extremely LT-tolerant varieties like Tohoku 234 maintained a stable panicle architecture. RNA sequencing of the early panicles revealed LT-induced expression of stress response genes in all varieties. Compared with Hitomebore and Sasanishiki, in Tohoku 234, the expression of genes involved in flowering and sugar metabolism—such as *OsGI* and *OsTOC1*—showed stepwise induction with decreasing temperatures, while the expression of genes related to the cell cycle exhibited stepwise suppression. In addition, 24 genes with variety-specific expression patterns were identified. These findings suggested that LTs during the early reproductive stage increased spikelet numbers, along with total anther numbers, which may reduce the pollen formation capacity within each anther in LT-susceptible varieties. This study offers insights into rice’s LT tolerance mechanisms.

## 1. Introduction

Extreme weather events have been increasingly reported worldwide as a consequence of global warming, resulting in reduced yields of major crops [1,2,3,4]. Rice is native to the tropical and subtropical regions of Southeast Asia and Africa, where it typically thrives under high-temperature and humid conditions but is highly sensitive to LTs. As a result, unusually localized cold spells resulting from global warming continue to cause significant LT injuries in rice, with substantial yield losses in various regions across the world [5,6].

The study and breeding of rice that is resistant to LT injury have a long history. The period with the highest susceptibility to LT damage is during the booting stage, during which injuries lead to abnormal enlargement of the tapetum and hinder pollen maturation in anthers [7,8]. Moreover, in strains that are sensitive to LT, there is a distinct tendency for these conditions to lead to a decrease in anther length and pollen numbers. Quantitative trait loci (QTLs) linked to cold tolerance during the booting stage have been identified on all chromosomes, with at least eight key genes—*qCtb1*, *qCTB4a*, *OsbZIP73*, *qCTB4-2*, *qPSR10*, *OsLTT1*, *OsWRKY53*, and *qCTB7* [9,10,11,12,13,14,15,16]. The *qCtb1* gene encodes an F-box protein that participates in the cold response mechanism through the ubiquitin–proteasome pathway [9]. The receptor-like cytoplasmic kinase gene *qCTB4a* enhances fertility by interacting with mitochondrial ATP synthase b to maintain ATP levels during cold stress [10]. The transcription factor *OsbZIP73* promotes sugar transport from the anthers to pollen grains by suppressing abscisic acid accumulation and reactive oxygen species (ROS) levels [12]. The *OsLTT1* gene activates ROS metabolism to facilitate tapetum degradation under LT stress [13]. *qCTB4-2* boosts sterol glycosides to maintain cell membrane permeability and safeguard pollen grains, working synergistically with *qCTB4a* [10,14]. *OsWRKY53* is involved in the repression of gibberellin biosynthesis and negatively affects cold tolerance [15]. *qCTB7* on LOC_Os07g07690 encodes a PHD finger protein that modulates pollen structure and anther development under LT stress [16]. These mechanisms ultimately ensure the maintenance of anther size, the number of mature pollen grains, and the fertilization ability of pollen, thereby improving the seed setting rate and leading to increased yield.

Furthermore, the temperature during the early panicle formation stage has been identified as a factor influencing LT injury during pollen development [7,8,17]. High temperatures during the juvenile stage enhance tolerance to LTs during the subsequent risk period, whereas LTs increase susceptibility to LT injury [17,18]. However, the mechanism by which temperature variation during the early panicle formation stage intensifies LT injury remains unclear. Rice’s panicle architecture, which encompasses the branching patterns and overall structure of rice inflorescences, is significantly influenced by environmental factors such as temperature, nutrient availability, drought, and high ozone levels [19,20,21]. These environmental factors can influence the number and length of branches and ultimately affect grain productivity.

For many years, our Furukawa Agricultural Experiment Station has employed a unique LT experimental field equipped with a deep-water irrigation system to breed rice varieties with enhanced cold tolerance (Figure 1). In 1991, Tohoku 143 (later named Hitomebore) was developed, which inherited cold resistance traits from its parent, Koshihikari rice [22,23]. During the Great Cold Disaster of 1993 in Japan, Hitomebore demonstrated resistance to LTs, whereas Sasanishiki, a cultivar that had been dominant in the Tohoku region until then, was severely damaged [24]. Since then, Hitomebore has been widely cultivated throughout this region. In recent years, we developed Tohoku 234, a variety that is extremely LT- and high-temperature-tolerant [25] and which maintains a high fertilization rate, even at an LT of 18.5 °C. In contrast, the fertilization rate of Hitomebore declined significantly at this temperature. In this study, we investigated LT tolerance, panicle architectural changes, and gene expression variation during early panicle development in rice varieties grown in LT paddy fields at 19.0 °C and 18.5 °C [26] compared with natural field conditions. Detailed analyses of gene expression and changes in panicle architecture were conducted using three varieties: Tohoku 234, Hitomebore, and Sasanishiki. In addition, for 25 varieties (including the parent and grandparent lines of Tohoku 234) that have nearly the same heading date as these 3 varieties under natural conditions but differ in terms of cold tolerance, we evaluated the relationship between seed fertility and changes in panicle architecture (number of spikelets per panicle length) under three different temperature conditions.

## 2. Results

### 2.1. Panicle Architecture Changes in Response to LT Stress

The 28 *japonica* varieties used in this study were classified into four groups based on their seed fertility under LT treatment conditions of 18.5 °C: extremely strong (average seed fertility of 70% or more), including Tohoku 234; strong (average seed fertility of 50 to less than 70%), including Furukawa Taireichubo 111 and Tohoku 187, the parent and grandparent of Tohoku 234; moderate (average seed fertility of 30–50%), including Hitomebore; and weak (average seed fertility of less than 30%), including Sasanishiki (Figure 2, Appendix A). For the extremely strong and strong varieties, seed fertility remained largely unchanged between 19.0 °C and 18.5 °C conditions. In contrast, the moderate varieties exhibited significantly lower fertility at 18.5 °C than at 19.0 °C. The weak varieties had already experienced a notable decline in fertility at 19.0 °C, which further worsened at 18.5 °C.

The number of panicles per plant was reduced by half at 19.0 °C and leveled off at 18.5 °C in all 28 varieties compared with the natural ambient temperature (NT) (Figure 3a, Appendix A). The reduction in tiller number likely played a significant role in this outcome, as it was caused by limiting the cultivation area per plant in the LT experimental field to approximately one-third of that in the natural ambient-temperature field (see Section 4). In contrast to the marked changes in panicle number per plant, the panicle length showed a more gradual tendency to decrease under LT conditions. A slight reduction at 19.0 °C and a plateau at 18.5 °C were observed across all 28 varieties (Figure 3b, Appendix A). Moreover, the culm length was largely unaffected by the LT treatment (Appendix A).

In contrast, almost all varieties showed a decreasing trend in total spikelet number per panicle at 19.0 °C, whereas the extremely strong and strong varieties remained almost unchanged at 18.5 °C, and the moderate and weak varieties showed a reverse increasing trend at 18.5 °C (Figure 4a, Appendix A). The spikelet numbers per panicle and fertility rate are plotted in Figure 4b. We found little correlation between these factors.

Furthermore, when plotting the spikelet number per panicle length, we observed that most of the extremely strong and strong varieties tended to decline at 19.0 °C and leveled off at 18.5 °C, whereas the medium varieties tended to decline at 19.0 °C but reversed at 18.5 °C, and most of the weak varieties increased at 18.5 °C (Figure 5a, Appendix A). Notably, this factor of spikelet number per panicle length showed a higher correlation with the fertility rate due to LTs, which was more pronounced at 18.5 °C than at 19.0 °C (Figure 5b). In other words, even in varieties with moderate LT tolerance, such as Hitomebore, Furukawa Taireichubo 111 (Furukawa 111), Akita PL1, and Norin 24, the seed setting rate decreased significantly under the more severe LT conditions of 18.5 °C. Additionally, under these harsh LT conditions, the number per panicle length increased markedly (Figure 5b, black ●). This trend was similar to that observed when fertility declined in the varieties with weak LT tolerance.

Since there was a correlation between the four different cold-tolerant groups, particularly in the spikelet number per panicle length and fertility rates, we next observed changes in the panicle architecture due to LTs in Tohoku 234, Hitomebore, and Sasanishiki, using these as representatives of three groups.

Figure 6a shows the typical panicle architecture of each variety grown under NT, 19.0 °C, and 18.5 °C conditions. Figure 6b shows enlarged images of the top three apical branches and bottom three basal branches in each panicle at different temperatures. The number of spikelets in the apical and basal branches of each variety was plotted (Figure 6c). The results showed that in the LT-tolerant Tohoku 234 variety, the overall panicle shape became more compact under LT conditions, and the panicle architecture (such as spikelet numbers in each branch) tended to remain almost unchanged under severe 18.5 °C conditions. In contrast, in the LT-sensitive Sasanishiki, a marked decrease in the spikelet number on the apical branches and a marked increase in the number of spikelets on the lower basal branches were observed in response to LT conditions. The same trend was observed in the moderately tolerant Hitomebore, but the change was slightly less pronounced at 19.0 °C. These results showed that when LTs led to an increase in spikelet density on the basal branches, it caused injury that significantly reduced seed fertility. Conversely, in the extremely strong varieties, changes in spikelet density were notably suppressed.

### 2.2. Transcriptional Changes in Early Panicles in Three Varieties Grown at Different Temperatures

To investigate the relationship between differences in LT tolerance and changes in panicle architecture, we analyzed the gene expressions of the rice varieties Tohoku 234, Hitomebore, and Sasanishiki during the early developmental stage (when young panicles were approximately 1.0 cm in length) using RNA sequencing. Appendix A shows a heatmap clustering of the top 100 genes with the most variable expression levels, based on 27 RNA sequencing datasets from three varieties, three temperature conditions, and three biological replicates. In addition, Appendix A shows the results of the PCA of expression data for all 44,823 genes, using the mean values from three biological replicates. These analyses revealed that variations in gene expression were influenced by both differences among varieties and cultivation temperature conditions.

Similarly to the other two varieties, although Tohoku 234 showed little architectural change in the panicle due to LTs, our PCA revealed that even a small temperature difference of just 0.5 °C, between 19.0 °C and 18.5 °C, caused significant changes in gene expression compared with natural temperatures (Appendix A). To visualize the genes with differential expression in Tohoku 234, we used MA plots and a Venn diagram (Figure 7a–c). At 19.0 °C, approximately 400 genes showed up- or downregulation in response to cold, whereas at 18.5 °C, both expression changes increased in over 1000 genes. GO enrichment analysis of these differentially expressed genes revealed that under LT, genes related to the “Response to stimulus” were predominantly and notably up- and downregulated in a temperature-dependent manner (Figure 7d,e). The GO term “Developmental process” was also enriched in up- and downregulated genes. “Response to hormone” was enriched in upregulated genes at LT. In contrast, GO terms such as “Protein metabolic process” and “Response to stimulus” were enriched in downregulated genes at both 19 °C and 18.5 °C, and “Cell cycle” only at 18.5 °C.

Next, for the GO-enriched genes in Tohoku 234 that showed expression changes in response to LTs, we analyzed the gene expressions in Hitomebore and Sasanishiki using RNA sequencing data from young panicles. At this sampling stage, the expression of *OsACO3*, a gene involved in ethylene biosynthesis [27], increased under LT conditions in all varieties (Figure 8). Several cold-responsive genes, such as *OsASR1*, *OsASR3*, *OsDR8*, and *OsNAC6*, were upregulated in each variety at LTs. In addition, the heat stress transcription factors *HsfA3* and *HsfC1b*, as well as the HSP DnaJ *OsDjC28* and the small HSP Os10g0159600 gene, were also induced in each variety. Conversely, the expressions of *OsCAL1*, *OsAPX2*, and *OsROLE1*, which are involved in different stress response pathways as documented by Oryzabase [28], were reduced at LTs.

Furthermore, the *OsGI* (GIGANTEA) gene, which is involved in flowering time control, circadian clock regulation, sugar metabolism, and stress tolerance [29], was upregulated in each variety at LTs (Figure 8). Pseudo-response regulator (*PRR*) genes belonging to the same GO category, such as *OsTOC1* (*PRR1*), *OsPRR59*, *OsPRR95*, and the F-box protein *OsFKF1* gene, were also induced. Moreover, among these genes, the stepwise induction of expression from the NT to LTs was more prominent in Tohoku 234 than in the other two cultivars. In contrast, the circadian clock-associated *CCA1* and *OsMADS50* genes were suppressed under LT conditions. Table 1 shows the sowing, transplanting, and heading dates of the three varieties. Under normal temperature conditions in the main field, the heading dates were almost the same for all three varieties, occurring approximately four months after sowing. Under the LT condition of 19.0 °C, Tohoku 234’s heading was delayed by 10 days, whereas those of the other two cultivars were delayed by 16 days. At 18.5 °C, Tohoku 234’s heading was delayed by an additional day, and the other two cultivars were delayed by an additional three days. Among the varieties, the delay in heading due to LTs was the shortest for Tohoku 234. In addition, some regulatory genes involved in cell cycle progression and proliferation, such as *OsCDC48E*, *OsCycD5;3*, *OsKRP4*, and *OsKRP*5, were suppressed, and this downregulation was more significant in Tohoku 234.

The quantitative real-time PCR-based gene expression analysis results—using three additional independent biological replicates for the four genes associated with stress response and flowering time regulation (*OsDR8*, *OsGI*, *OsTOC1*, and *OsCCA1*), as depicted in Figure 9—was similarly aligned with those obtained from the RNA sequencing analysis.

Moreover, analysis of the RNA-seq expression data from Tohoku 234, Hitomebore, and Sasanishiki intriguingly revealed 24 variety-specific genes that were expressed in one or two of these varieties. These genes were spread across chromosomes 1, 4, 5, 6, 7, 8, 9, 10, and 11 (Figure 10). Eight of these twenty-four genes showed specific expression in Tohoku 234, with one gene each on chromosomes 4 and 6, two on chromosome 5, and four on chromosome 8. Among these genes, only Os06g0254300, an ortholog of *Arabidopsis thaliana* caleosin 4 (*AtCLO4*), a negative regulator of abscisic acid signaling [30], and *OsFBX286* have been annotated. Conversely, Os01g0103400, Os05g0132800, Os07g0159200, Os09g0325300, and Os11g0623700 were only expressed in Sasanishiki. Only two genes were specifically expressed in Hitomebore, while five were expressed in both Hitomebore and Sasanishiki, and the remaining four were expressed in both Tohoku 234 and Hitomebore.

## 3. Discussion

The architecture of rice panicles, which significantly affects the grain yield, is regulated by various genes and influenced by numerous environmental factors [31,32]. Spikelet density is a crucial trait that affects both the yield and quality of grains, making its optimization essential for maximizing grain production [32]. There is a well-documented trade-off between the number of grains per panicle and the grain filling [33]. While increasing the number of grains per panicle can enhance the yield potential, it may also lead to smaller, less filled grains, particularly in inferior spikelets. This is because a greater number of grains can intensify competition for resources, such as photosynthates and nutrients, which may affect grain weight and, ultimately, yield. In rice, originally grown in tropical and subtropical regions, LT environments significantly reduce metabolic activity, including photosynthetic activity. In addition, LT injury in rice primarily results from the abortion of pollen development and leads to male sterility [7,8,34]. In other words, during the inflorescence development stage, which determines the final panicle architecture, even at an LT of approximately 19.0 °C, where cold damage occurs, a certain number of spikelets still develop, waiting for a later recovery in temperature. However, if LTs persist, even though the development and fertility of the female ovule are maintained, the male pollen formation becomes incomplete.

In this study, we investigated the changes in panicle architecture under LT stress conditions of 19.0 °C and 18.5 °C, which cause cold injury to rice during the reproductive growth stage (from inflorescence development to heading). We examined the differences among newly bred varieties with extreme tolerance (extremely strong), strong tolerance (strong), tolerance (moderate), and susceptibility (weak). Interestingly, weak varieties showed a marked tendency for an increase in spikelet number per panicle length in response to LTs. In contrast, in moderately tolerant varieties, the number decreased at 19.0 °C; however, under more severe 18.5 °C conditions, where seed fertility was significantly reduced, there was a pronounced increase in the number of spikelets, similar to that in the weak varieties. In contrast, in the extremely strong and strong varieties, the number decreased at 19.0 °C and remained low, even at 18.5 °C (Figure 5a). In other words, an inverse correlation was revealed between spikelet number per panicle length and cold tolerance (seed setting rate) under LT conditions (Figure 5b).

Furthermore, we found that in Sasanishiki, a cultivar that is conventionally sensitive to cold, the number of spikelets on the primary basal branches increased stepwise in response to lower temperatures, such as 19.0 °C and 18.5 °C (Figure 6). In contrast, the number of spikelets on the upper branches decreased by LTs, resulting in noticeable architectural changes in the panicle. Similar changes were observed in varieties with moderate LT tolerance, such as Hitomebore, which became more pronounced at a lower temperature of 18.5 °C. In contrast, in varieties with extremely strong LT tolerance, such as Tohoku 234, the basal and upper primary branch lengths became shorter, but changes in the number of spikelets were minimal, and structural changes in the panicle due to LTs were relatively mild (see Figure 11).

Even in Tohoku 234, which showed robustness to changes in panicle architecture caused by LTs, our analysis of the gene expression in the early inflorescent panicles revealed that in response to LTs of 19.0 °C and 18.5 °C, the expressions of ethylene synthesis genes, stress-related genes that are responsive to cold and abscisic acid, and molecular chaperones were also upregulated in Hitomebore and Sasanishiki (Figure 7 and Figure 8). In contrast, the expressions of genes such as *OsAPX2* and *OsCAL1*, which are involved in reactive oxygen species, cadmium, and diseases [28], decreased at LTs, which was particularly pronounced in Tohoku 234 (Figure 8). This may be because various stress response genes were activated, thereby enhancing stress resistance and detoxification functions, which in turn may have resulted in lower levels of ROS and other stress molecules than usual.

Furthermore, in this study, we found that the expression of genes involved in regulating flowering time exhibited various changes in response to LTs. The expression of the *OsMADS50* transcription factor, which, when overexpressed, induces early flowering and when suppressed by RNAi or knockout results in delayed flowering [35,36], was significantly reduced by LTs in all three varieties. This phenomenon is thought to explain the delay in the heading period owing to LTs, as shown in Table 1. In contrast, the expression levels of *OsGI* and *OsFKF1*, which also positively regulate flowering, as well as *OsTOC1*, *OsPRR59*, and *OsPRR95*, which act as repressors in the clock transcription–translation feedback loop in *Arabidopsis* [37], were all increased in each early panicle by LTs (Figure 8). The induction of expression was greater in Tohoku 234 than in the other two cultivars, which may be related to the shorter delay in heading caused by LTs in this variety. In addition, the expressions of some cell cycle regulators, such as *OsCDC48*, *OsCycD5;3*, *OsKRP4*, and *OsKRP5*, were predominantly suppressed in Tohoku 234 under LT conditions (Figure 8). This could be linked to the lack of an increase in spikelets at the basal branches under LT conditions.

In conclusion, this study shows that abnormalities in pollen formation due to LT stress in rice are linked to changes in panicle architecture during inflorescence development. It was clarified that in conventionally tolerant and susceptible varieties, where LTs lead to reduced seed fertility, the number of spikelets per panicle length increases, a phenomenon that is particularly evident in the basal branches of the panicle. Conversely, extremely strong LT-tolerant varieties bred in recent years, such as Tohoku 234, have developed mechanisms to maintain their panicle architecture as much as possible without increasing the number of spikelets per panicle length. They achieve this while diversely regulating the expression of stress response and stress adaptation genes, as well as genes involved in flowering time and cell division control, to the same or greater extent than conventional varieties. Furthermore, comprehensive gene expression analysis identified variety-specific candidate loci (Figure 10), providing valuable resources for future functional validation of these genes and for investigating loci that are involved in LT tolerance, including the surrounding genes.

In this study, we focused exclusively on gene expression analysis at the early young panicle stage, a critical phase for determining panicle architecture. However, a comprehensive understanding of LT tolerance requires analyses of entire reproductive stages and tissues, such as anthers during the booting stage, as well as vegetative organs like roots and stems, which are also affected by deep-water cooling treatment. Furthermore, data from the RiceXPro database indicated that at least 18 of the 24 variety-specific genes are widely expressed across all tissues and organs of rice, including roots, leaves, panicles, and anthers. Therefore, further investigation into gene expression changes under LT conditions beyond the early panicle stage is essential. As climate change continues to challenge crop production, deepening our understanding of how plants integrate environmental signals into reproductive growth programs is essential for the development of high-yield and stress-tolerant cultivars.

## 4. Materials and Methods

### 4.1. Plant Materials and Growth Conditions

In this study, 28 (*Oryza sativa* L.) *japonica* rice varieties, including Sasanishiki, Hitomebore, Tohoku 234, its parent Furukawa Taireichubo 111, and grandparent Tohoku 187 (Appendix A), were examined, all of which have nearly identical heading dates under natural ambient-temperature field conditions. LT stress was applied using a deep-water irrigation system (19.0 °C and 18.5 °C, 25 cm deep; Figure 1) from the primordial stage to the completion of heading (approximately 2 months, from early July to early September) at the Miyagi Prefectural Furukawa Agricultural Experimental Station (38°36.1′ N, 140°54.7′ E). For the control plants, the average maximum and minimum temperatures were 30.7 °C and 22.4 °C, respectively, from early July to early September 2024.

In both the natural-temperature and cold-water paddies, two seedlings were planted in each hole, with row spacings of 30 cm and 24 cm and plant spacings of 22 cm and 10 cm, respectively, in the middle of May (Table 1). In cold-water paddy fields at 19.0 °C and 18.5 °C, around 100 plants from each of the three varieties were cultivated for detailed analysis, including gene expression studies. For the remaining 25 varieties, three plants per variety were grown in two separate replicates (six plants each) with different cultivation plots. The number of panicles per plant was measured. Additionally, for each six plants x top five panicles = 30 panicles per biological replicate (*n* = 29 for T187, 18.5 °C), we investigated the seed setting rate, spikelet number, and spikelet number per panicle length at the NT, 19.0 °C, and 18.5 °C. Basal fertilizers were applied at rates of 0.3–0.4, 0.45–0.6, and 0.35–0.47 kg a^−1^ (nitrogen, PO_5_, and potassium) 5–7 days before transplanting. The panicle architecture and seed setting rate of each variety were monitored after grain maturation. All plant experiments described in this study were performed according to the relevant institutional, national, and international guidelines and legislation.

### 4.2. Expression Analyses in Juvenile Panicles with RNA Sequencing

Young panicles, each about 1.0 cm long, were collected from the experimental fields under natural ambient, 19.0 °C, and 18.5 °C temperature conditions for Tohoku 234, Hitomebore, and Sasanishiki varieties. The samples were carefully dissected using plain forceps and immediately stored in a dry ice box. Total RNA was extracted from 5 frozen panicles per sample using the RNeasy Plant Mini Kit (QIAGEN, Hilden, Germany), following the manufacturer’s protocol. Three biological replicates were used for RNA sequencing analysis. The total RNA concentration was assessed using a Synergy LX microplate reader (BioTek, Winooski, VT, USA) with the QuantiFluor RNA System (Promega, Fitchburg, WI, USA), followed by integrity evaluation with a 5200 Fragment Analyzer System and Agilent High Sensitivity RNA Kit (Agilent Technologies, Santa Clara, CA, USA). Strand-specific libraries were prepared using the MGIEasy RNA Directional Library Prep Set (MGI Tech, Shenzhen, China) according to the manufacturer’s protocol. Library concentrations were determined using a Synergy H1 microplate reader (BioTek) with the QuantiFluor dsDNA system (Promega). Library qualities were confirmed using an Agilent 2100 BioAnalyzer and a Fragment Analyzer (Advanced Analytical Technologies, Cypress, CA, USA). Circularized DNA molecules were generated using the MGIEasy Circularization Kit (MGI Tech), and DNA nanoballs were prepared using the DNBSEQ-G400RS High-throughput Sequencing Kit (MGI Tech). Paired-end (2 × 100 bp) sequencing was conducted on a DBBSEQ-G400 platform. Adapters were trimmed using Cutadapt v4.0 [38], and low-quality reads were removed with Sickle v1.33 [39]. Filtered reads were aligned to RAP-DB transcript sequences (Nipponbare IRGSP-1.0; version 2021, 44,823 isoforms) using Hisat2 v2.2.1 [40]. Alignment files were processed with SAMtools v1.16.1 [41] and quantified at the gene level using featureCounts v2.0.0 [42], and expression values were normalized using the TPM method [43].

### 4.3. Quantitative RT-PCR Analysis

For qRT-PCR analysis, RNA was extracted from each frozen panicle (approximately 1.0 cm in length) with TRIzol Reagent (Invitrogen, Waltham, MA, USA). First-strand cDNAs were synthesized using the PrimeScript RT reagent Kit with DNA Eraser (TaKaRa, Shiga, Japan) according to the manufacturer’s protocol. Quantitative real-time RT-PCR was performed using the TB Green Premix Ex Taq II (Tli RNaseHPlus; TaKaRa) on a CFX96 thermal cycler (BioRad, Hercules, CA, USA). In the varieties Tohoku 234, Hitomebore, and Sasanishiki, the expression levels of the following stress response- and flowering-related genes were examined: Os07g00529600 (*OsDR8*), Os01g0182600 (*OsGI*), Os02g0618200 (*OsTOC1*), and Os08g0157600 (*OsCCA1*). Expression levels were normalized to Os01g0328400 (*OsUBQ5*) as the internal reference gene. All analyses were performed with at least three biological replicates, each with three technical replicates per sample. Gene-specific primers used are listed in Appendix A.

### 4.4. Statistical Analysis and Data Visualization

All statistical analyses and data visualizations were performed using R (v4.4.1, 2024) [44] within the RStudio (2023.6.1.524) environment [45] and Microsoft Excel 365 [46], unless otherwise specified. For all agronomic traits measured, data are summarized as mean ± standard deviation (SD). Statistical analyses were conducted using one-way analysis of variance (ANOVA) to determine significant differences within and between varieties and treatments, followed by the Tukey–Kramer (Appendix A) and least significant difference (LSD) (Figure 6c, Figure 8, Figure 9 and Figure 10) post hoc tests at *p* < 0.05.

### 4.5. Differentially Expressed Genes (DEGs)

Expression values were normalized using edgeR’s (v4.0) Trimmed Mean of M-values (TMM) method [47], which adjusts for differences in library size. Gene expression analyses comprised data from three biological replicates of RNA sequencing data. DEGs were identified using the edgeR (v4.0) package [47] in R, and the expression values of the LT treatments (19.0 °C and 18.5 °C, respectively) were compared with those of NT conditions in each variety. Significant DEGs were selected with FDR < 0.05 and log_2_ (count per million) > 0. DEGs were visualized using MA plots in Microsoft Excel 365 [46]. InteractiVenn [48], a web-based tool for set analysis using Venn diagrams, was used for correlation analysis between the DEG sets.

### 4.6. Functional Categorization of Genes

Gene ontology (GO) analyses were performed using g: Profiler [49], an R web-based toolset. Each analysis was performed with the statistical domain scope set to all known genes, a significance threshold based on the Benjamini–Hochberg FDR, and the data source restricted to the Biological Process category. The enriched GO terms of interest were selected based on their relevance to this study.

### 4.7. Evaluation of Panicle Architecture and Yield-Related Traits

At maturity, six plants were sampled, and the top five panicles from each plant were obtained (*n* = 30 per biological replicate for all varieties from the NT and LT field experiments at 19.0 °C and 18.5 °C), except for Tohoku 187 at 18.5 °C, where 29 panicles were used. Fertility rate was calculated as the percentage of filled spikelets relative to the total number of spikelets per panicle. Culm length (*n* = 6 plants), panicle number per plant (*n* = 6 plants), and panicle length (*n* = 6 panicles each from 6 independent plants) were measured. The spikelet number per panicle length was calculated by dividing the average spikelet number (*n* = 30 each, 29 for Tohoku 187 at 18.5 °C) by the average panicle length. For Tohoku 234, Hitomebore, and Sasanishiki varieties, representative images of the panicles, including three apical and three basal branches, were captured using a digital camera (EOS Kiss X9; Canon, Tokyo, Japan). Additionally, the spikelet number on the three apical and three basal branches of five representative panicles was counted manually.

## Figures and Tables

**Figure 1 plants-14-02759-f001:**
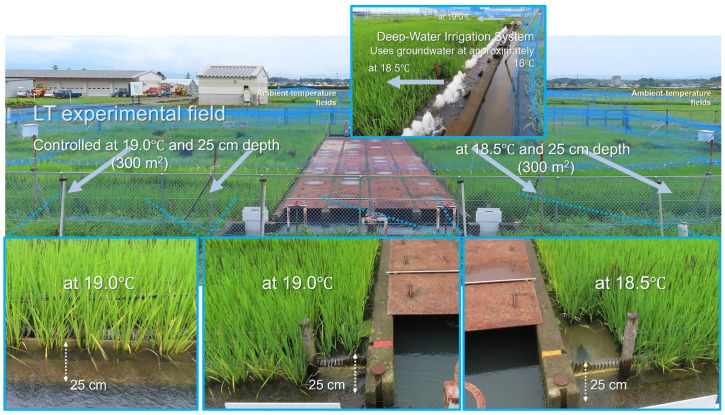
LT experimental fields, maintained at 19.0 °C and 18.5 °C using a deep-water irrigation system, alongside natural ambient-temperature paddy fields at the Miyagi Prefectural Furukawa Agricultural Experimental Station. Each year, more than 1000 breeding varieties and cultivars are tested for LT tolerance. See Section 4 for cultivation conditions.

**Figure 2 plants-14-02759-f002:**
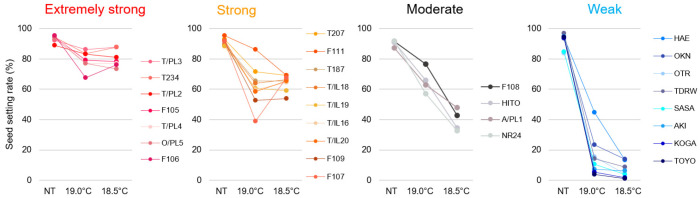
Effects of LTs on seed fertility in 28 *japonica* rice varieties. Rice plants were cultivated under natural ambient-temperature (NT) and LT conditions, as described in the Materials and Methods Section (Section 4.1). Seed setting rates are shown as percentages of (mean ± standard deviation, SD) fertile spikelets relative to the total number of spikelets per panicle at maturity. Six plants × top 5 panicles = 30 panicles per biological replicate (except for T187 at 18.5 °C, where *n* = 29). One-way analysis of variance (ANOVA), followed by the Tukey–Kramer post hoc test, was performed to assess significant differences between temperature conditions and among varieties under NT (see Appendix A). Variety abbreviations: T/PL3, Tohoku PL3; T234, Tohoku 234; T/PL2, Tohoku PL2; F105, Furukawa 105; T/PL4, Tohoku PL4; T/PL5, Tohoku PL5; F106, Furukawa 106; T207, Tohoku 207; F111, Furukawa 111; T187, Tohoku 187; T/IL18; Tohoku IL18; T/IL19, Tohoku IL19; TIL16, Tohoku IL16; T/IL20, Tohoku IL20; F109, Furukawa 109; F107, Furukawa 107; F108, Furukawa 108; HITO, Hitomebore; A/PL1, Akita PL1; NR24, Norin 24; HAE, Haenuki; OKN, Okiniiri; OTR, Ootori; TDRW, Todorokiwase; SASA, Sasanishiki; AKI, Akihomare; KOGA, Koganehikari; TOYO, Toyonishiki.

**Figure 3 plants-14-02759-f003:**
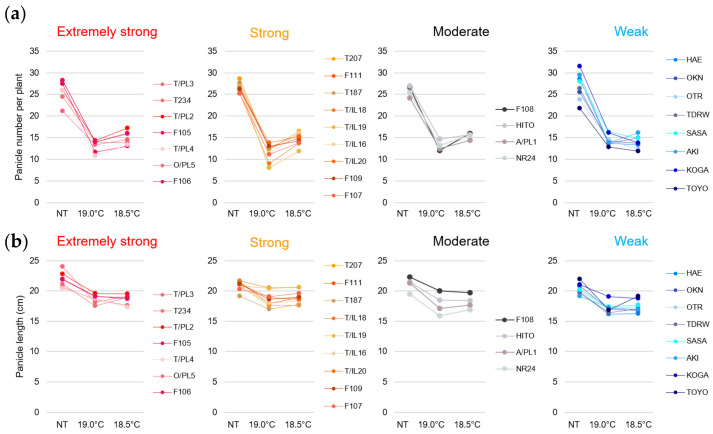
Effects of LTs on panicle number per plant and panicle length in 28 rice varieties. Rice plants were cultivated under NT and LT conditions as described in the Materials and Methods Section (Section 4.1). (**a**) Number of panicles per plant (*n* = 6). (**b**) Panicle length (main culm; *n* = 6). Values are shown as mean ± SD. One-way ANOVA, followed by the Tukey–Kramer post hoc test, was performed to evaluate significant differences between varieties under NT and between temperature conditions within each variety (see Appendix A).

**Figure 4 plants-14-02759-f004:**
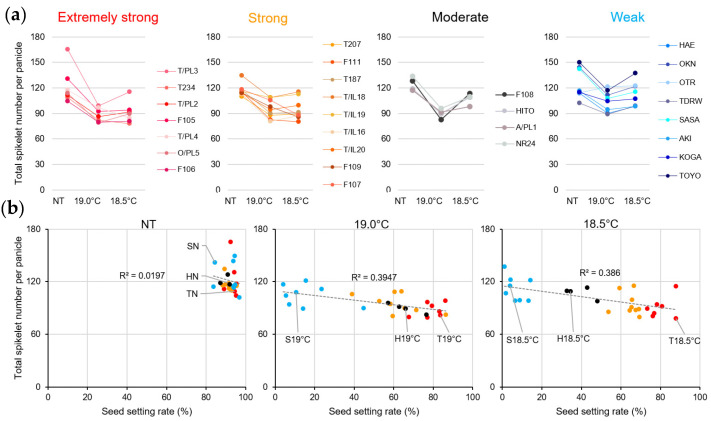
Effect of LTs on the total spikelet number per panicle in 28 rice varieties. Rice plants were cultivated under NT and LT conditions as described in the Materials and Methods Section (Section 4.1). (**a**) Total spikelet number per panicle. Six plants × top 5 panicles = 30 panicles per biological replicate (except for T187 at 18.5 °C, where *n* = 29). (**b**) Correlation of total spikelet number per panicle with seed setting rates under NT (left panel), 19 °C (middle panel), and 18.5 °C (right panel). Tohoku 234 at NT (TN), 19.0 °C (T19 °C), and 18.5 °C (T18.5 °C). Hitomebore at NT (HN), 19.0 °C (H19 °C), and 18.5 °C (H18.5 °C). Sasanishiki at NT (SN), 19.0 °C (S19 °C), and 18.5 °C (S18.5 °C). Red dots: extremely strong varieties, Orange dots: strong varieties, Black dots: moderate varieties, and Blue dots: weak varieties. Values are shown as the mean ± SD. One-way ANOVA, followed by the Tukey–Kramer post hoc test, was performed to evaluate significant differences between varieties under NT and between temperature conditions within each variety (see Appendix A).

**Figure 5 plants-14-02759-f005:**
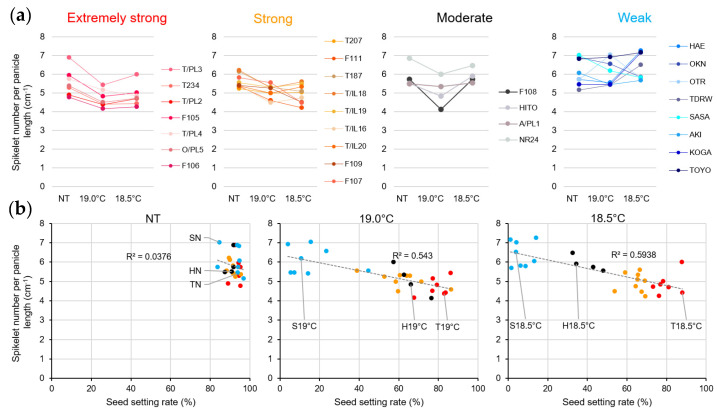
Effects of LTs on spikelet density in 28 rice varieties. Rice plants were cultivated under NT and LT conditions as described in the Materials and Methods Section (Section 4.1). (**a**) Spikelet number per panicle length. Calculated from dividing the total spikelet numbers (*n* = 30 panicles; 29 were used for T187 at 18.5 °C) by the average value of panicle length. (**b**) Correlation between spikelet number per panicle length and seed setting rates under NT (left panel), 19 °C (middle panel), and 18.5 °C (right panel). Red dots: extremely strong varieties, Orange dots: strong varieties, Black dots: moderate varieties, and Blue dots: weak varieties. Values are shown as the mean ± SD. One-way ANOVA, followed by the Tukey–Kramer post hoc test, was performed to assess significant differences between temperature conditions and among varieties under NT (see Appendix A).

**Figure 6 plants-14-02759-f006:**
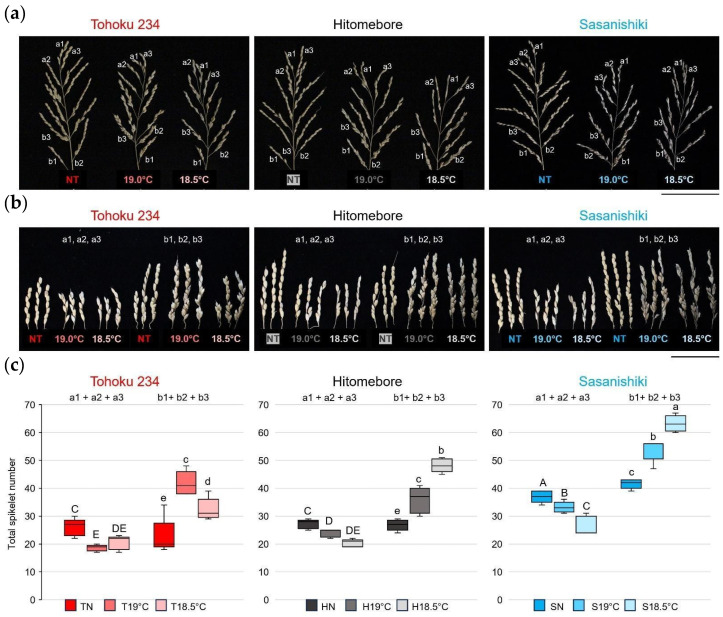
Effects of LT stress on panicle architecture in Tohoku 234, Hitomebore, and Sasanishiki. Rice varieties were cultivated under NT and LT conditions as described in the Materials and Methods Section (Section 4.1). (**a**) Typical mature panicles at harvest. Scale bar: 10.0 cm. (**b**) Representative first (a1), second (a2), and third apical (a3) and basal (b1, b2, and b3) branches at harvest are shown. Scale bar: 5.0 cm. (**c**) Total spikelet numbers of the three (1 + 2 + 3) apical and basal branches. *n* = 5 panicles per biological replicate. Significant differences are indicated by different uppercase letters for apical branches and lowercase letters for basal branches (*p* < 0.05, one-way ANOVA with Fisher’s least significant difference [LSD] test).

**Figure 7 plants-14-02759-f007:**
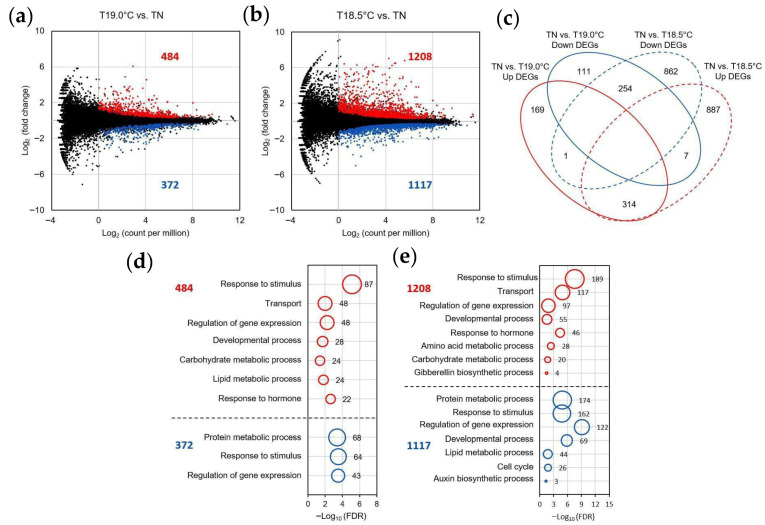
LT-induced gene expression dynamics in young panicles of the Tohoku 234 rice varieties. Gene expression analysis was performed using three biological replicates of young panicles grown under NT and LT conditions as described in the Materials and Methods Section (Section 4.1). (**a**,**b**) MA plots of differentially expressed genes (DEGs) between LT (19 °C and 18.5 °C) and NT conditions. DEGs were identified using the following criteria: FDR < 0.05 and log_2_ (count per million) > 0. Red dots represent upregulated DEGs; sapphire blue represents downregulated DEGs. (**c**) Venn diagram showing overlaps between DEGs identified in 19.0 °C vs. NT and 18.5 °C vs. NT comparisons. Up, upregulated; down, downregulated. (**d**,**e**). Bubble plot representations of gene ontology (GO) enrichment for up- and downregulated DEGs at 19.0 °C and 18.5 °C, respectively (g: Profiler; FDR < 0.05; Biological Process). The *x*-axis represents the −log_10_(FDR), the *y*-axis represents the GO term names, and the bubble size reflects the number of genes associated with each term.

**Figure 8 plants-14-02759-f008:**
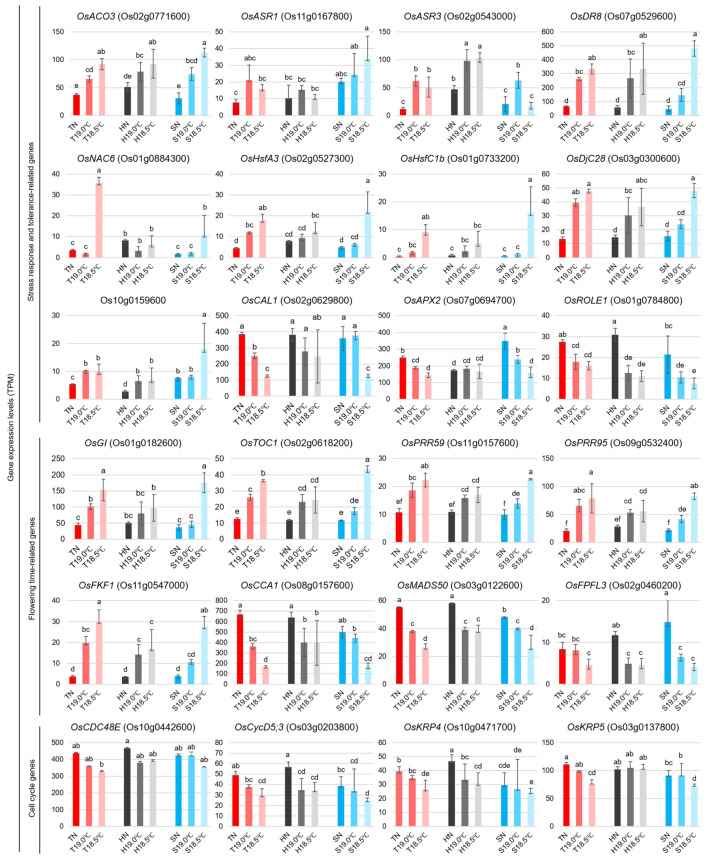
TPM-normalized expression levels of 24 typical genes that change under different temperatures. Genes were selected from the “Response to stimulus” GO term in Figure 7d,e and were further grouped into two functional classes: stress response- and tolerance-related genes, and flowering time-related genes. Additional genes associated with cell cycle regulation were selected from the “Cell cycle” GO term in Figure 7e. The expression values represent the mean ± SD of three biological replicates of Tohoku 234 compared with those in Hitomebore and Sasanishiki. Tohoku 234 at NT (TN), 19.0 °C (T19 °C), and 18.5 °C (T18.5 °C). Hitomebore at NT (HN), 19.0 °C (H19 °C), and 18.5 °C (H18.5 °C). Sasanishiki at NT (SN), 19.0 °C (S19 °C), and 18.5 °C (S18.5 °C). Bar graphs display the mean ± SD of TPM-normalized expression levels from three biological replicates. Statistical significance was represented with different letters based on one-way ANOVA and the LSD test.

**Figure 9 plants-14-02759-f009:**
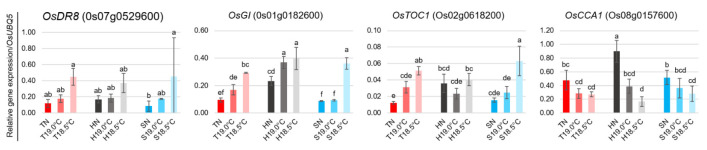
Relative expression of 4 genes related to stress response and flowering time regulation from Figure 8. qRT-PCR was performed using gene-specific primers, and the expression ratios were normalized to *OsUBQ5*. Data represent three biological replicates, each with three technical replicates, and are presented as the mean ± SD. Different letters above bars indicate significant differences (*p* < 0.05; ANOVA, followed by LSD post hoc test).

**Figure 10 plants-14-02759-f010:**
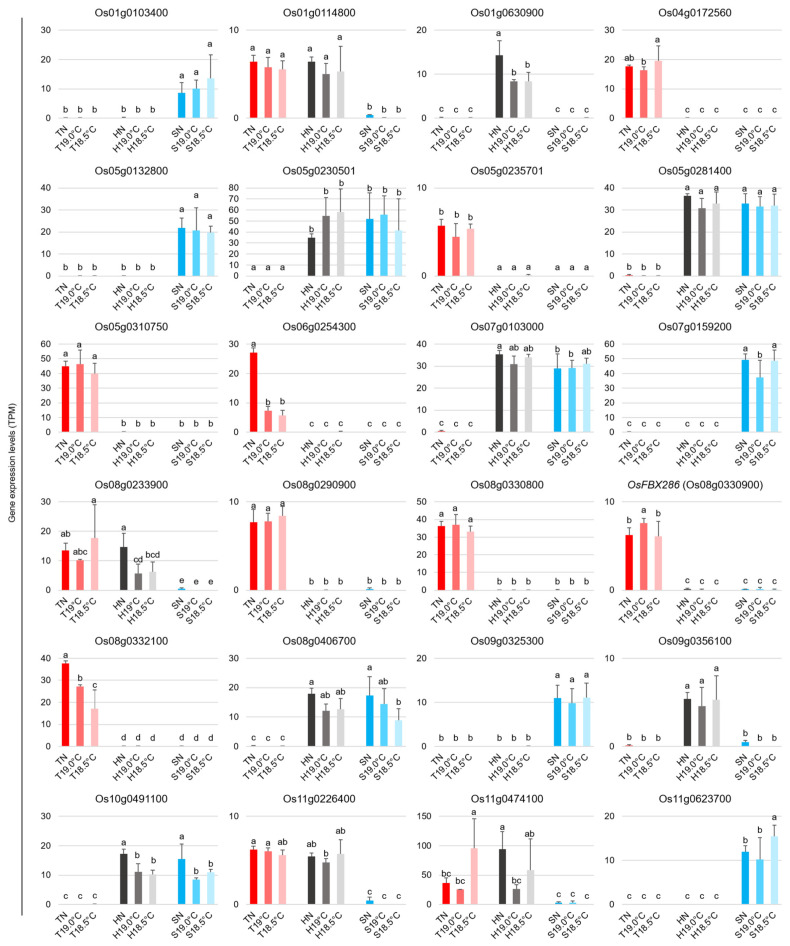
TPM-normalized expression levels of 24 genes with variety-specific expression patterns. Genes were selected based on their expression in only one or two of these varieties, irrespective of temperature conditions. Bar graphs represent the mean ± SD of TPM-normalized expression levels from three biological replicates. Statistical significance is represented with different letters based on one-way ANOVA and the LSD test.

**Figure 11 plants-14-02759-f011:**
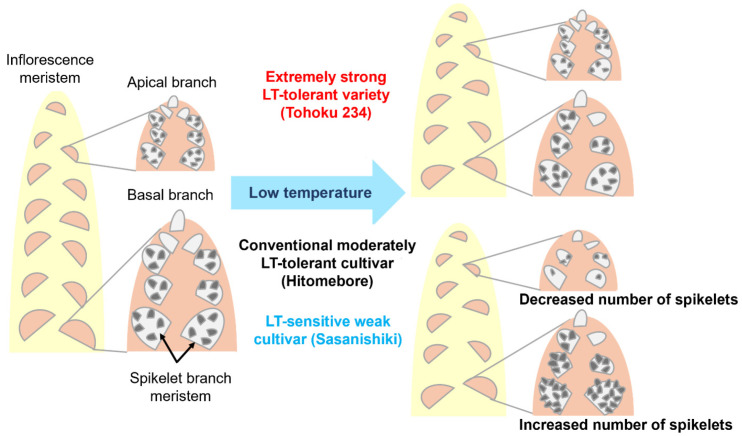
Illustration summarizing the relationship between extremely strong tolerance and robustness of panicle architecture.

**Table 1 plants-14-02759-t001:** Sowing, transplanting, and heading dates of Tohoku 234, Hitomebore, and Sasanishiki.

Study Area	Variety Name	Sowing Date	Transplanting Date	Heading Date
Ambient-condition	Tohoku 234	16 April	22 May	5 August
paddy field (NT)	Hitomebore	16 April	22 May	4 August
	Sasanishiki	16 April	22 May	5 August
Cold-water	Tohoku 234	16 April	14 May	15 August
paddy field (19.0 °C)	Hitomebore	16 April	14 May	20 August
	Sasanishiki	16 April	14 May	21 August
Cold-water	Tohoku 234	16 April	14 May	16 August
paddy field (18.5 °C)	Hitomebore	16 April	14 May	23 August
	Sasanishiki	16 April	14 May	24 August

Note 1: Fertilization conditions in the test area (basal fertilizer: N-top fertilizer N kg/a): Main ambient paddy field (0.3–0.0), cold-water (deep-water irrigation system) paddy (0.4–0.0). Note 2: Planting density: main ambient paddy field—15.2 plants/m^2^, single planting; cold-water paddy—41.7 plants/m^2^, double planting. Note 3: Cold-water treatment: The water temperature was maintained at 19.0 °C (or 18.5 °C). The plots were irrigated with circulating water at a depth of about 15 cm from 26 June to 12 July and at a depth of about 25 cm from 13 July to 4 September in 2024.

## Data Availability

Data are contained within the article and Appendix A. The global gene expression datasets have been deposited in the BioProject database under the accession number [PRJDB35721].

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
