# Peer review of "Low-Temperature-Induced Changes in Rice Panicle Architectures and Their Robustness in Extremely Cold-Tolerant Cultivars"

_plants, 2025, doi:10.3390/plants14172759_

Round 1

Reviewer 1 Report

Comments and Suggestions for Authors

As strange as it may seem to people who are not familiar with rice plants, during the period of global warming rice suffers from low temperatures. Particularly severe damage can be caused if the cold snap occurs during the development of pollen, which is very sensitive to low temperatures. In order to understand possible mechanisms of protection against cold snap, the authors carried out a very interesting study. They studied the effect of cold stress (19.0 °C and 18.5 °C) during the reproductive growth stage on the panicle architecture and fertility of 28 rice varieties with different levels of cold tolerance. Carefully performed studies led the authors to an interesting and important conclusion. It turned out that rice varieties with low cold tolerance had an increased spikelet density on the basal branches, and the most resistant varieties (for example, Tohoku 234) retained a stable panicle architecture. RNA sequencing revealed some specificities in gene expression in early panicles in varieties with different cold tolerance. Not surprisingly, stress response genes are expressed in response to stress in all studied varieties. Some specificity in gene expression was found for the most cold-tolerant rice varieties. In contrast to the opinion of the authors, the molecular part of the work, in the opinion of the reviewer, does not offer new insights into the mechanisms of cold tolerance.

Thus, the work is interesting, its first part is done extremely carefully. RNA sequencing yielded some important results. It is possible that, based on the variability of transcriptomes, it would be very useful in the future to create and analyze additional transcriptomes for other stages of panicle formation. What I like most is that the nature of panicle formation in different rice varieties can be used to judge the cold tolerance of plants.

There are two small comments.

  1. Without any doubts about the quality of this work, it would be very useful to present the main characteristics of the quality of the transcriptomes in Suppl. materials.
  2. It is known that the transcriptome is a very complex multi-stage approach for analyzing the level of transcripts. A mandatory stage of the work is the validation of the obtained results. As the authors know very well, for this purpose take 8-12 genes, determine the level of their transcripts under the conditions of creating transcriptomes using the qRT-PCR method and compare them with the transcriptome data. If the obtained data are consistent, then the transcriptome results can be considered correct. It is possible that in the future this requirement will be canceled, but for now it is better to provide such control.

I hope that the authors will not have any difficulties in making the noted corrections to this manuscript.

Author Response

Thank you very much for taking the time to improve our study. We hope that the revised manuscript is suitable for publication.

Comment 1. Without any doubts about the quality of this work, it would be very useful to present the main characteristics of the quality of the transcriptomes in Suppl. materials.

Yes, according to your valuable comment, we have added Supplementary Figure 2ab, which includes (1) a heatmap of the top 100 genes showing the most variable expression differences among all RNA sequencing data, and (2) a PCA analysis of the nine experimental groups based on the average normalized log2CPM values from the three biological replicates in each group. These results demonstrate differences in gene expression among the three varieties and further confirm the effects of cultivation temperature. This content has now been newly described in the main text on lines 193–204.

"Figure S2a showed a heatmap clustering of the top 100 genes with the most variable expression levels, based on 27 RNA sequencing datasets from three cultivars, three temperature conditions, and three biological replicates. In addition, Figure S2b showed the results of the PCA analysis of expression data for all 44,823 genes, using the mean values from three biological replicates. These analyses revealed that variations in gene expression were influenced by both differences among varieties and cultivation temperature conditions.

Similar to the other two varieties, although Tohoku 234 showed little architectural change in the panicle due to LT, PCA revealed that even a small temperature difference of just 0.5°C, between 19.0°C and 18.5°C, caused significant changes in gene expression compared to natural temperatures (Figure S2b). To visualize the genes with differential expression in Tohoku 234, we used MA plots and a Venn diagram (Figure 7a-c). "

Comment 2. It is known that the transcriptome is a very complex multi-stage approach for analyzing the level of transcripts. A mandatory stage of the work is the validation of the obtained results. As the authors know very well, for this purpose take 8-12 genes, determine the level of their transcripts under the conditions of creating transcriptomes using the qRT-PCR method and compare them with the transcriptome data.

Yes, in this revision, we conducted qRT-PCR analysis with three new biological replicates, focusing particularly on the gene expression of OsGI, OsTOC1, OsCCA1, and OsDR8, which are involved in flowering timing control, sugar metabolism, and stress response. The results shown in a new Figure 9 reproduced the results obtained from the RNA sequencing (Figure 8). This content has now been newly described in the main text on lines 263–266.

"The quantitative real-time PCR-based gene expression analysis results, using three additional independent biological replicates for the four genes associated with stress response and flowering time regulation—OsDR8, OsGI, OsTOC1, and OsCCA1—depicted in Figure 9, similarly aligned with those obtained from the RNA sequencing analysis."

We sincerely thank you once again for your valuable comment.

Reviewer 2 Report

Comments and Suggestions for Authors

review of Plants-3807689

Low temperature-induced changes in rice panicle architecture and their robustness in extreme cold-tolerant cultivars

Masato Kisara, Aisha Ahmad Abu, and Atsushi Higashitani

The authors studied the effects of “low temperature” (by rice standards) on panicle development in 28 rice cultivars varying in their cold tolerance. They identified 4 groups that varied from very tolerant of LT to very sensitive. They then focused on 3 cultivars that varied in LT-tolerance and made detailed comparisons of their panicle architecture and gene expression under ambient and low temperature conditions. They conclude that LT- sensitive cultivars increase the number of spikelets on the basal branches under LT stress whereas LT resistant cultivars maintain stable panicle architecture. They also identified genes that were differentially expressed in  LT-sensitive and LT-resistant cultivars under LT that may explain these differences.

Overall, the study appears to have been conducted with suitable techniques, adequate replication and analysis, with one critical problem. What temperatures were the panicles experiencing? It appears that the roots were being treated with chilled water, but how did that affect the temperature at the panicles? I do not question that they indeed identified significant differences in response to paddy water temperature.  However, is this a direct response to the temperatures experienced by the panicles themselves, or an indirect response to signals coming from the roots? In this context, does 18.5 ˚C vs 19.0 ˚C paddy water have any effect on the temperatures that the panicles experience?

The English is good, but there are numerous minor mistakes. Most are related to tense (past, present, future) and singular versus plural.  I have listed some below, but there are many others.

Line 20: please change “strong” to “LT-tolerant”

Line 55: please change “several” to “many”

Lines 67 -69: please rewrite for clarity

Line 85: please delete the second  “of”

Lines 121-124: please rewrite for clarity

Lines 146-147: please rewrite for clarity

Lines 160-161: please rewrite for clarity

Lines 212-214: please rewrite for clarity

Lines 230 and 235: please state the reference gene used to quantitate expression and how differential expression was determined.

Line 335:  Please delete “were applied.”

Lines 355- 360: did you analyze all of your libraries using all 4 of these instruments/kits?

Line 401: please change “from which” to “and”

Lines 409-410: please change to “dividing the average spikelet number by the average panicle length (N = 30 each, 29 for Tohoku 187 at 18.5 °C).”

Comments on the Quality of English Language

The English is good, but there are numerous minor mistakes. Most are related to tense (past, present, future) and singular versus plural.  I have listed some below, but there are many others.

Author Response

Thank you very much for taking the time to improve our study. We hope that the revised manuscript is suitable for publication.

Major Comment 1. Overall, the study appears to have been conducted with suitable techniques, adequate replication and analysis, with one critical problem. What temperatures were the panicles experiencing? It appears that the roots were being treated with chilled water, but how did that affect the temperature at the panicles? I do not question that they indeed identified significant differences in response to paddy water temperature.  However, is this a direct response to the temperatures experienced by the panicles themselves, or an indirect response to signals coming from the roots? In this context, does 18.5 ˚C vs 19.0 ˚C paddy water have any effect on the temperatures that the panicles experience?

Indeed, as you mentioned, we have yet to ascertain whether the response is directly experienced by the panicle itself or indirectly originates from the roots or stem. In this study, we confirmed gene expression in rice panicles at a stage where reproductive growth initiates at the base and gradually advances upward as differentiation occurs, specifically when the panicle reaches about 1 cm in length. At this stage, the sampled section is entirely submerged under the 25 cm deep cold-water irrigation condition.

We are particularly interested in the difference between 19.0°C and 18.5°C. In rice varieties susceptible to low temperatures, we observed that this 0.5°C difference had a more pronounced effect on panicle architecture. Conversely, traits such as culm length and panicle number did not exhibit significant differences between the 19.0°C and 18.5°C cultivation conditions in any of the varieties, leading us to suspect that the response is relatively specific to the panicle itself. However, we cannot draw a definitive conclusion at this point.

Therefore, in this revised version, we have included descriptions of the observed changes in traits such as culm length and panicle number (lines 108-117). As you suggested, I have also noted at the end of the Discussion that it will be important in the future to conduct comprehensive analyses of gene expression in various tissues and at different developmental stages, including anthers, to account for possible influences from the roots or stem (lines 378-386).

Lines 108-117:

"The number of panicles per plant was reduced by half at 19.0°C and leveled off at 18.5°C in all 28 varieties compared to the natural ambient temperature (NT) (Figure 3a, Table S1). The reduction in tiller number likely played a significant role in this outcome, as it was caused by limiting the cultivation area per plant in the LT experimental field to approximately one-third of that in the natural ambient-temperature field (see Materials and Methods). In contrast to the marked changes in panicle number per plant, panicle length showed a more gradual tendency to decrease under LT conditions. A slight reduction at 19.0°C and a plateau at 18.5°C were observed across all 28 varieties (Figure 3b, Table S1). Moreover, culm length was largely unaffected by the LT treatment (Table S1)."

Lines 378-386:

"In this study, we focused exclusively on gene expression analysis at the early young panicle stage, a critical phase for determining panicle architecture. However, a comprehensive understanding of LT tolerance requires analyses of entire reproductive stages and tissues, such as anthers during the booting stage, as well as vegetative organs like roots and stems, which are also affected by deep-water cooling treatment. Furthermore, data from the RiceXPro database indicated that at least 18 of the 24 variety-specific genes are widely expressed across all tissues and organs of rice, including roots, leaves, panicles, and anthers. Therefore, further investigation into gene expression changes under LT conditions is essential beyond the early panicle stage. "

The English is good, but there are numerous minor mistakes. Most are related to tense (past, present, future) and singular versus plural.  I have listed some below, but there are many others.

Thank you very much for your corrections. 

Line 20: please change “strong” to “LT-tolerant”

Yes, we have changed. Thank you.

Line 55: please change “several” to “many”

Yes, we did.

Lines 67 -69: please rewrite for clarity

Yes, we have rewritten this sentence below.

Lines 85-90: "In addition, for 25 varieties (including the parent and grandparent lines of Tohoku 234) that have nearly the same heading date as these three varieties under natural conditions but differ in cold tolerance, we evaluated the relationship between seed fertility and changes in panicle architecture (number of spikelets per panicle length) under three different temperature conditions."

Line 85: please delete the second  “of”

Yes, we did.

Lines 121-124: please rewrite for clarity

Yes, we have rephrased this sentence more clearly as follows:

Lines 152-157: "In other words, even in varieties with moderate LT tolerance, such as Hitomebore, Furukawa Taireichubo 111 (Furukawa 111), Akita PL1, and Norin 24, the seed setting rate significantly decreased under the more severe LT conditions of 18.5°C. Additionally, under these harsh LT conditions, the number per panicle length increased markedly (Figure 5b, black ●). This trend was similar to that observed when fertility declined in the varieties with weak LT tolerance."

Lines 146-147: please rewrite for clarity

Thank you, we have rewritten it.

Lines 179-182: "These results showed that when LT led to an increase in spikelet density on the basal branches, it caused injury that significantly reduced seed fertility. Conversely, in the extremely strong varieties, changes in spikelet density were notably suppressed."

Lines 160-161: please rewrite for clarity

Yes, we have changed this sentence more clearly as follows:

Lines 193-204: "Figure S2a showed a heatmap clustering of the top 100 genes with the most variable expression levels, based on 27 RNA sequencing datasets from three varieties, three temperature conditions, and three biological replicates. In addition, Figure S2b showed the results of the PCA analysis of expression data for all 44,823 genes, using the mean values from three biological replicates. These analyses revealed that variations in gene expression were influenced by both differences among varieties and cultivation temperature conditions.

Similar to the other two varieties, although Tohoku 234 showed little architectural change in the panicle due to LT, PCA revealed that even a small temperature difference of just 0.5°C, between 19.0°C and 18.5°C, caused significant changes in gene expression compared to natural temperatures (Figure S2b). To visualize the genes with differential expression in Tohoku 234, we used MA plots and a Venn diagram (Figure 7a-c). "

Lines 212-214: please rewrite for clarity

Thank you very much. We have rewritten it.

Lines 272-274: "Moreover, analysis of the RNA-seq expression data from Tohoku 234, Hitomebore, and Sasanishiki intriguingly revealed 24 variety-specific genes that were expressed in one or two of these varieties. "

Lines 230 and 235: please state the reference gene used to quantitate expression and how differential expression was determined.

These analyses were the results of RNA sequencing with three biological replicates, and the normalized TPM values were plotted. Therefore, there is no reference value like qRT-PCR results. We have clearly rewritten this.

Lines 239-240: "Bar graphs display the mean ± SD of TPM-normalized expression levels from three biological replicates. Statistical significance was represented with different letters based on one-way ANOVA and the LSD test. "

Lines 285-286: "Bar graphs represent the mean ± SD of TPM-normalized expression levels from three biological replicates. Statistical significance was represented with different letters based on one-way ANOVA and the LSD test."

Line 335:  Please delete “were applied.”

Yes, thank you.

Lines 355- 360: did you analyze all of your libraries using all 4 of these instruments/kits?

Yes, we did adjust the library in four steps, but the text wasn't clear, so we rewrote it.

Lines 422-426: "Strand-specific libraries were prepared using the MGIEasy RNA Directional Library Prep Set (MGI Tech) according to the manufacturer’s protocol. Library concentrations were determined using a Synergy H1 microplate reader (BioTek) with the QuantiFluor dsDNA system (Promega). Library qualities were confirmed using an Agilent 2100 BioAnalyzer and a Fragment Analyzer (Advanced Analytical Technologies). "

Line 401: please change “from which” to “and”

Yes, we changed it. Thanks.

Lines 409-410: please change to “dividing the average spikelet number by the average panicle length (N = 30 each, 29 for Tohoku 187 at 18.5 °C).”

Yes, we did.

We have also carefully corrected the verb tenses. Thank you very much for your kind review.

Reviewer 3 Report

Comments and Suggestions for Authors

This manuscript presents data on low-temperature induced changes in rice panicle architecture. Certain things on the research design and methods are essential but are not clear.

Introduction is very short and should include more references on rice tolerance to low temperature stress. In addition, please give more information on the cultivar Tohoku 234 which is tolerant to high temperatures and low temperature?

In Materials and Methods, the research design is not clear, therefore I can not fully judge the merit of this research study. There is no information on the experimental design.

Please discuss in detail  what kind of experimental design did you use for the 28 rice cultivars that were grown in the field.

How many replications did you use? How did you plant the seeds, in row plots?

How many rows and what was the plant density you used? How many plants did you use for sampling?

For total RNA extraction, how many young panicles did you use per cultivar?

Did you only plant the experiment in one location and one year?

Furthermore, I believe that the authors should add a picture that shows the conventionally tolerant and susceptible cultivars, where the number of spikelets per panicle length increases, a phenomenon particularly evident in the basal branches of the panicle, comparted to extremely strong LT-tolerant cultivars bred in recent years, such as Tohoku 234, that have developed a mechanism to maintain panicle architecture without increasing the number of spikelets per panicle length.

This photograph will strengthen their conclusion.

Author Response

Thank you very much for taking the time to improve our study. We hope that the revised manuscript is suitable for publication.

Comment 1: Introduction is very short and should include more references on rice tolerance to low temperature stress. In addition, please give more information on the cultivar Tohoku 234 which is tolerant to high temperatures and low temperature?

Yes, it's a very important suggestion. Based on your suggestion, I added some information about QTL analysis and target genes in the introduction (Lines 42-60).

On the other hand, this is the first research paper on the characters of Tohoku 234, and the only one reference [25] is an academic proceeding in Japanese.

Lines 42-60: "Moreover, in strains sensitive to LT, there was a distinct tendency for these conditions to lead to a decrease in anther length and pollen numbers. Quantitative trait loci (QTLs) linked to cold tolerance during the booting stage have been identified on all chromosomes, with at least eight key genes, including qCtb1, qCTB4a, OsbZIP73, qCTB4-2, qPSR10, OsLTT1, OsWRKY53, and qCTB7 [9-16]. The qCtb1 gene encodes an F-box protein that participates in the cold response mechanism through the ubiquitin-proteasome pathway [9]. The receptor-like cytoplasmic kinase gene qCTB4a enhances fertility by interacting with mitochondrial ATP synthase b to maintain ATP levels during cold stress [10]. The transcription factor OsbZIP73 promotes sugar transport from anthers to pollen grains by suppressing abscisic acid accumulation and reactive oxygen species (ROS) levels [12]. The OsLTT1 gene activates ROS metabolism to facilitate tapetum degradation under LT stress [13]. qCTB4-2 boot sterol glycosides to maintain cell membrane permeability and safeguard pollen grains, working synergistically with CTB4a [10,14]. OsWRKY53 is involved in the repression of gibberellin biosynthesis and negatively affects cold tolerance [15]. qCTB7 on LOC_Os07g07690 encodes a PHD finger protein that modulates pollen structure and anther development under LT stress [16]. These mechanisms ultimately ensure the maintenance of anther size, the number of mature pollen grains, and the fertilization ability of the pollen, thereby improving the seed setting rate and leading to increased yield."

Comment 2: In Materials and Methods, the research design is not clear, therefore I can not fully judge the merit of this research study. There is no information on the experimental design.

Please discuss in detail  what kind of experimental design did you use for the 28 rice cultivars that were grown in the field.

How many replications did you use? How did you plant the seeds, in row plots?

How many rows and what was the plant density you used? How many plants did you use for sampling?

Thank you for your accurate comments. We have added a new Figure 1 of the cultivation method and have provided more details in Materials and Methods Section.

Lines 91-94: "Figure 1 LT experimental fields maintained at 19.0°C and 18.5°C using a Deep-Water Irrigation System, alongside natural ambient-temperature paddy fields at the Miyagi Prefectural Furukawa Agricultural Experimental Station. Each year, more than 1,000 breeding varieties and cultivars have been tested for LT tolerance. See the Materials and Methods section for cultivation conditions.  

Lines 392-406: "In this study, 28 rice (Oryza sativa L.) japonica varieties, including Sasanishiki, Hitomebore, Tohoku 234, its parent Furukawa Taireichubo 111, and grandparent Tohoku 187 (Table S1), were examined, all of which have nearly identical heading dates under natural ambient-temperature field conditions. LT stress was applied using a deep-water irrigation system (19.0°C and 18.5°C, 25 cm deep; Figure 1) from the primordial stage to the completion of heading (approximately 2 months, from early July to early September) at the Miyagi Prefectural Furukawa Agricultural Experimental Station (38°36.1′N, 140°54.7′E). For the control plants, the average maximum and minimum temperatures were 30.7°C and 22.4°C, respectively, from early July to early September 2024. In both the natural temperature and cold-water paddies, two seedlings were planted in each, with row spacings of 30 cm and 24 cm and plant spacings of 22 cm and 10 cm, respectively, in the middle of May (Table 1). In cold-water paddy fields at 19.0℃ and 18.5℃, around 100 plants from each of the three varieties were cultivated for detailed analysis, including gene expression studies. For the remaining 25 varieties, three plants per variety were grown in two separate replicates with different cultivation plots." 

Lines 475-482: "At maturity, six plants were randomly sampled, and the top five panicles from each plant were obtained; (n=30 per biological replicate for all varieties from the NT and LT (19.0°C and 18.5°C) field experiments, except for Tohoku 187 at 18.5°C, where 29 panicles were used. Fertility rate was calculated as the percentage of filled spikelets relative to the total number of spikelets per panicle. Culm length (n=6 plants), panicle number per plant (n=6 plants), and panicle length (n=6 panicles each from 6 independent plants) were measured. Spikelet number per panicle length was calculated by dividing the average spikelet number (n=30 each, 29 for Tohoku 187 at 18.5 °C) by the average panicle length. "

Comment 3: For total RNA extraction, how many young panicles did you use per cultivar?

We added the following sentence in Materials and Methods section.

Lines 413-418: "Young panicles, each about 1.0 cm long, were collected from the experimental fields under natural ambient, 19.0°C and 18.5°C temperature conditions for Tohoku 234, Hitomebore, and Sasanishiki varieties. The samples were carefully dissected using plain forceps and immediately stored in a dry ice box. Total RNA was extracted from 5 frozen panicles per sample using the RNeasy Plant Mini Kit (QIAGEN), following the manufacturer’s protocol. Three biological replicates were used for RNA sequencing analysis."

Comment 4: Did you only plant the experiment in one location and one year?

Yes. All samples were cultivated in 2024 season in LT experimental fields maintained at 19.0°C and 18.5°C using a Deep-Water Irrigation System and natural ambient-temperature paddy fields at the Miyagi Prefectural Furukawa Agricultural Experimental Station (see new Figure 1).

We sincerely thank you once again for your valuable comment.

Comment 5: Furthermore, I believe that the authors should add a picture that shows the conventionally tolerant and susceptible cultivars, where the number of spikelets per panicle length increases, a phenomenon particularly evident in the basal branches of the panicle, comparted to extremely strong LT-tolerant cultivars bred in recent years, such as Tohoku 234, that have developed a mechanism to maintain panicle architecture without increasing the number of spikelets per panicle length.

This photograph will strengthen their conclusion.

Thank you for your excellent suggestion. However, we are unable to present photographs of the panicle morphology of the other 25 lines. These 25 varieties were cultivated in 3 individual plants x 2 rows (see Lines 403-406), and all were used for seed fertility studies. Hence, it is not possible to take photographs of the entire panicle morphology now. The three representative varieties (Tohoku 234, Hitomebore, and Sasanishiki) were cultivated in approximately 100 individual plants each. 

Round 2

Reviewer 2 Report

Comments and Suggestions for Authors

review of Plants-3807689 version 2

Low temperature-induced changes in rice panicle architecture and their robustness in extreme cold-tolerant cultivars

Masato Kisara, Aisha Ahmad Abu, and Atsushi Higashitani

The authors studied the effects of “low temperature” (by rice standards) on panicle development in 28 rice cultivars varying in their cold tolerance. They identified 4 groups that varied from very tolerant of LT to very sensitive. They then focused on 3 cultivars that varied in LT-tolerance and made detailed comparisons of their panicle architecture and gene expression under ambient and low temperature conditions. They conclude that LT- sensitive cultivars increase the number of spikelets on the basal branches under LT stress whereas LT resistant cultivars maintain stable panicle architecture. They also identified genes that were differentially expressed in  LT-sensitive and LT-resistant cultivars under LT that may explain these differences.

The authors have satisfactorily addressed the concerns raised in my previous review.

I recommend that they state in their revision that when the panicle is about 1 cm long the sampled section is entirely submerged in the cold-water irrigation so that people like me who aren’t familiar with the details of rice cultivation will recognize that the developing panicles were in fact subjected to the indicated temperatures.

Two very minor points:

Line 60: please change “boot” to “boosts”

Line 150: Please start the caption with “Figure 3.”

Author Response

Thank you very much for your supports. I have changed all.

Reviewer 3 Report

Comments and Suggestions for Authors

Authors have made improvements in their manuscript but the fundamental problem remains. These are data that are derived from only one location with only 2 replications.

According to the authors, all samples were cultivated in 2024 season in LT experimental fields maintained at 19.0°C and 18.5°C using a Deep-Water Irrigation System and natural ambient-temperature paddy fields at the Miyagi Prefectural Furukawa Agricultural Experimental Station

In addition, the 25 varieties were cultivated in only 3 individual plants x 2 rows, and all were used for seed fertility studies, while the three representative varieties (Tohoku 234, Hitomebore, and Sasanishiki) were cultivated in approximately 100 individual plants each. 

Since these are only preliminary data, I suggest that the authors define the objective and innovation of their study and publish this research as a Note, not as an original article.

Furthermore, some sentences (especially in revisions) do not read well, English syntax needs improvement.

Comments on the Quality of English Language

English could be improved to more clearly express the research.

Author Response

Thank you very much for your comments and efforts. We have requested English editing services on the Plants journal website.

Furthermore, we used the top five panicles, a total of 30 panicles, for the analysis. As a response to your comment, we added the text " For the remaining 25 varieties, three plants per variety were grown in two separate replicates (six plants each) with different cultivation plots. The number of panicles per plant was measured. Additionally, for each six plants x top five panicles = 30 panicles per biological replicate (n = 29 for T187, 18.5°C), we investigated the seed setting rate, spikelet number, and spikelet number per panicle length at the NT, 19.0°C, and 18.5°C. " to the Materials and Methods Section in the 2nd revision.

This 2nd revision is suitable for publication in the Special Issue “Plant Functioning Under Abiotic Stress” of Plants.

Round 3

Reviewer 3 Report

Comments and Suggestions for Authors

The authors have not responded to all of my comments. 

1.These are data that are derived from only one location with only 2 replications, at the Miyagi Prefectural Furukawa Agricultural Experimental Station. With only one location and two replications, the analysis of data and conclusions can be ambiguous.

The effect of low-temperature-induced changes in rice panicle architectures and the investigation into gene expression changes under low-temperature conditions are essential traits that need more than one-location of data to make accurate predictions.

 2.The objective is not clear. I suggest that the authors define the objectives and innovation of their study and publish this research as a Note, not as an original article.

Author Response

Thank you very much for your efforts.

In your judgment, this study constitutes an important discovery, but further replication experiments are needed, and the current manuscript data is based on preliminary results.

(In typical plant science experiments, biological experiments using three more plants are also published in scientific journals. This study also conducted significance tests using 6 independent plants in each, including SD, to demonstrate significant differences between strains.)

However, we are unable to make further improvements, including further replication, at this time.

Therefore, we would appreciate it if you could instruct us to change the "Original Article" heading of the manuscript to "Communication" ("Note" is not available in "Plants"). This will allow us to publish more preliminary results in the current manuscript format.

On the other hand, if you could instruct us to delete all 25 lines of data and rewrite it when submitting to the journal, we will withdraw the submission.

We would appreciate your final decision and also editorial decision as soon as possible.

Thanks again for your kindness.

Best regards,

Atsushi